# Implant-to-implant wireless networking with metamaterial textiles

Xi Tian [1,2,11] ✉, Qihang Zeng[1,2,11], Selman A. Kurt [1,11], Renee R. Li[3,4], Dat T. Nguyen [1,2,5], Ze Xiong[1,2,6,7], Zhipeng Li [1], Xin Yang [1], Xiao Xiao[1], Changsheng Wu[2,6,8], Benjamin C. K. Tee [2,6,8], Denys Nikolayev [9], Christopher J. Charles [3,4,10] & John S. Ho [1,2,6] ✉

Implanted bioelectronic devices can form distributed networks capable of sensing health conditions and delivering therapy throughout the body. Current clinically-used approaches for wireless communication, however, do not support direct networking between implants because of signal losses from absorption and reflection by the body. As a result, existing examples of such networks rely on an external relay device that needs to be periodically recharged and constitutes a single point of failure. Here, we demonstrate direct implant-to-implant wireless networking at the scale of the human body using metamaterial textiles. The textiles facilitate non-radiative propagation of radio-frequency signals along the surface of the body, passively amplifying the received signal strength by more than three orders of magnitude (>30 dB) compared to without the textile. Using a porcine model, we demonstrate closed-loop control of the heart rate by wirelessly networking a loop recorder and a vagus nerve stimulator at more than 40 cm distance. Our work establishes a wireless technology to directly network body-integrated devices for precise and adaptive bioelectronic therapies.

Implanted bioelectronic devices such as stimulators[1–3], sensors[4–6], and drug delivery devices[7–9] have revolutionized the treatment of various disorders. Their miniaturization has led to the possibility of deploying multiple networked devices throughout the body, which can sense health conditions and deliver therapy to address a broad range of unmet clinical needs[10–12]. Distributed networks of bioelectronic implants can, as examples, adaptively regulate autonomic functions[13,14], provide closed-loop prosthetic sensory feedback[15], and autonomously manage diabetes[16]. However, the interconnection of distributed implanted devices into a functional network

remains a major technological challenge. Wireless networking approaches are necessary for the long-term operation of such implants[17,18]. Existing technologies used for wearable sensors[19,20], however, cannot be easily adapted for implantable devices due to losses resulting from signal reflection and absorption by biological tissues[21,22]. Consequently, existing examples of networked wireless implants all rely on indirect signal transmission mediated by a battery-powered relay device placed outside of the body[11,16]. Although this approach has some medical applications, its use outside of clinical settings is hindered by the need for the external

[1]Department of Electrical and Computer Engineering, National University of Singapore, Singapore 117583, Singapore. [2]Institute for Health Innovation and Technology, National University of Singapore, Singapore 117599, Singapore. [3]Cardiovascular Research Institute, National University Heart Centre, Singapore 117599, Singapore. [4]Department of Surgery, Yong Loo Lin School of Medicine, National University of Singapore, Singapore 119228, Singapore. [5]Integrative Sciences and Engineering Program, NUS Graduate School, National University of Singapore, Singapore 119077, Singapore. [6]The N.1 Institute for Health, National University of Singapore, Singapore 117456, Singapore. [7]Department of Biomedical Engineering, National University of Singapore, Singapore 117583, Singapore. [8]Department of Materials Science and Engineering, National University of Singapore, Singapore 117575, Singapore. [9]IETR – UMR 6164, CNRS, University of Rennes, Rennes, France. [10]Christchurch Heart Institute, Department of Medicine, University of Otago, Christchurch, New Zealand. [11]These authors contributed equally: Xi Tian, Qihang Zeng, Selman A. Kurt. ✉e-mail: tianxi@u.nus.edu; johnho@nus.edu.sg

device, which needs to be periodically recharged and constitutes a single point of failure[23–26].

Here, we demonstrate an approach to achieve direct implant-to-implant wireless networking of bioelectronic implants across the human body (Fig. 1a). Our approach employs wearable metamaterials—artificial materials with a subwavelength structure that enable extraordinary control over optical, acoustic, and radio-frequency fields[27,28]. Metamaterials have found broad applications in wireless technology, ranging from wireless communication[29,30] to remote sensing[31–33], wireless power transfer[34–36], and wave-based computing[37,38]. To implement our approach, we design metamaterial textiles that can passively facilitate the non-radiative propagation of radio-frequency signals along the surface of the body[39–41], enabling direct communication between implants using established wireless protocols (Fig. 1b, c). We establish wireless connectivity between the textile and implants by coherently amplifying the signal on the surface of the body using phased textile structures (Fig. 1d). These metamaterial textiles can be easily integrated into regular clothing[42] and are compatible with the Bluetooth Low Energy standard used in commercially available medical devices. We show that the wireless transmission efficiency between implants interconnected through such metamaterial textiles can be enhanced by over three orders of magnitude (>30 dB) compared to without the textile. Additionally, we demonstrate in a porcine model closed-loop heart rate control by wirelessly networking an implanted loop recorder and a vagus nerve stimulator (VNS) that are >2.5 cm deep and >40 cm apart.

## Results
### System overview
Efficient wireless implant networking is challenging due to the unique transmission characteristics of the human body at radio frequencies.

Unlike wearable devices that rely on wireless communication through the air, implants face significant communication challenges due to radio-frequency wave reflection and absorption by the heterogeneous biological tissues within the body, resulting in high transmission loss. To overcome these limitations, metamaterial textiles must meet three key criteria. Firstly, they must support surface modes over the operating frequency range, allowing wireless signals to propagate along the body contour and circumvent the need for direct transmission through tissue[40,43]. Secondly, the metamaterial must bridge the mismatch between air and tissue to efficiently convert the radiation from implants into surface waves. Finally, the metamaterial must be passive and wearable, such as by integrating them into conventional clothing.

We developed a metamaterial textile (Fig. 1a) that meets the requirements for direct implant-to-implant wireless networking in the human body. To illustrate its functionality, Fig. 1e demonstrates wireless signal transmission between an implanted sensor and a stimulator. The sensor, located 4 cm beneath the skin, emits a wireless signal via a dipole antenna parallel to the interface. The metamaterial structure is engineered to convert the wireless signal to highly confined surface waves that propagate along the textile. When the surface waves reach the end of the structure, they excite currents within a concentric ringed structure, which we refer to as a phased surface[34], that focuses the signal on the implanted receiver. Full-wave electromagnetic simulations show that the implant-to-implant transmission efficiency is over three orders of magnitude (>30 dB) higher than without the metamaterial textile for distances ranging from 10 to 50 cm (Fig. 1f). A comparison with a relay system that uses radiative communication devices placed directly over the sensor and stimulator shows an improvement in transmission efficiency of more than 20 dB (Fig. 1f). The subwavelength structuring of the metamaterial plays a

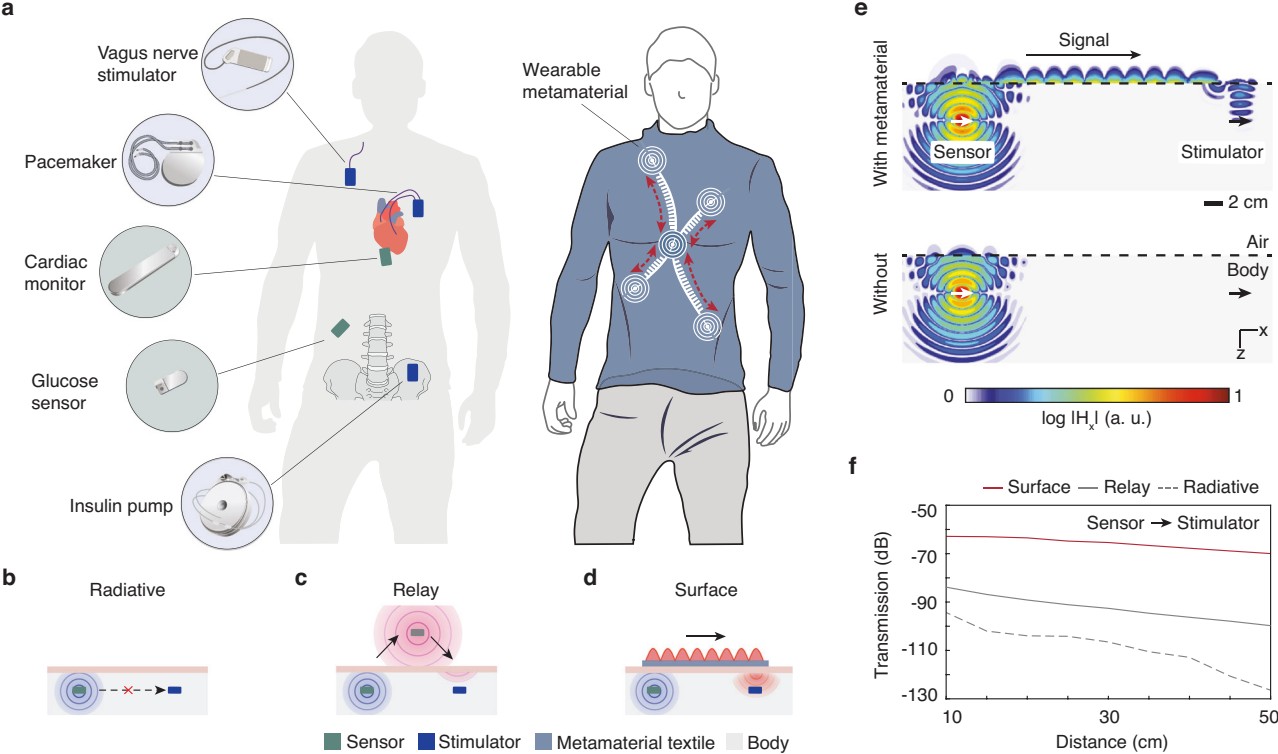

**Fig. 1 | Wireless networking of implantable devices with wearable metamaterials. a** Illustration of a wireless implantable medical devices network interconnected by metamaterial textiles. **b–d** Comparison of wireless implant-to-implant communication systems: **b** radiative interconnection is limited to transmission distance and implantation depth, **c** wireless relays cover longer distances but require additional transceivers, **d** metamaterial textiles enable efficient long-distance and deep-tissue transmission. **e** Simulated magnetic field distribution $|H_x|$ generated by a dipole implanted at a 4 cm depth with a metamaterial textile (top) and in the absence of metamaterial textile (bottom) above tissue. **f** Comparison of the transmission coefficient as a function of distance between the implanted antennas in (**e**).

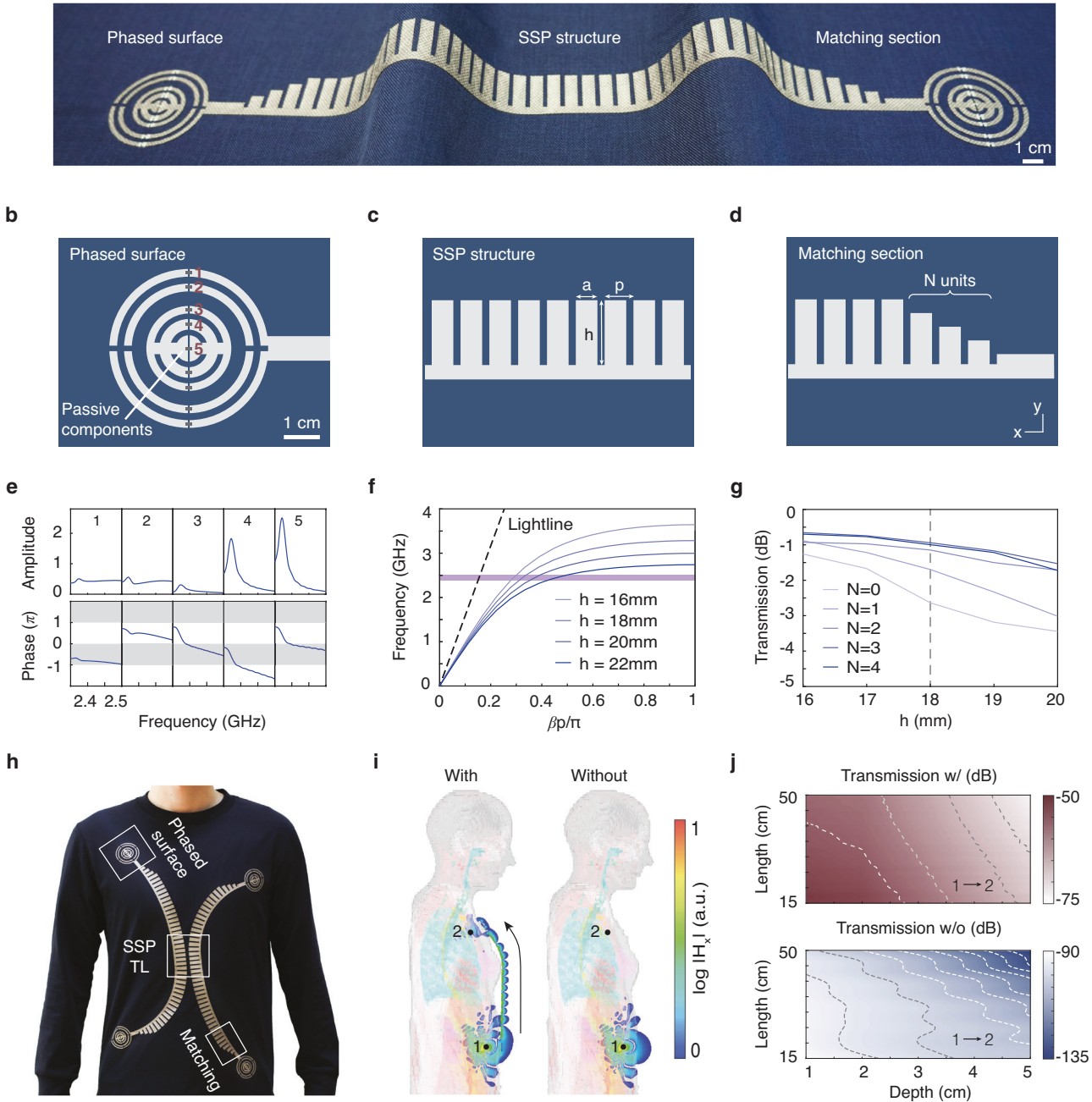

**Fig. 2 | Design and characterization of metamaterial textiles. a** Photograph of the metamaterial textile in a folded state. Scale bar, 1 cm. **b**–**d** Structure of metamaterial textiles including a phased surface loaded with passive elements (**b**), an SSP waveguide (**c**), and a matching section (**d**). **e** Amplitude and phase responses of the five ports labeled in (**b**) as a function of frequency. **f** Dispersion curves for SSP structure design with $h = 16$, 18, 20, and 22 mm, respectively. **g** Transmission coefficient as a function of $h$ for varying number of matching units ($N$). **h** Photograph of a metamaterial textile network integrating two SSP transmission lines (TL) and four phased surface terminals. **i** Comparisons of full-wave simulations of wireless implant-to-implant transmission in a computation human body model (28.95 dB enhancement with the metamaterial textile). Antennas are placed 4 cm below the skin (black dots). **j** Transmission coefficient with (top) and without (bottom) metamaterial textiles as a function of the length of the metamaterial textile and the depth of implantation, where dashed lines are contour lines for transmission levels with a spacing of 5 dB.

crucial role in achieving this enhanced transmission, as unstructured conductive textiles yield transmission efficiencies comparable to no textiles (Supplementary Figs. 1–3).

## System design

The metamaterial textile structure comprises a spoof surface plasmonic (SSP) waveguide[40], terminated by impedance matching sections[43] and phased surface structures[34] at both ends, as illustrated in Fig. 2a. Each component is made up of a conductive planar pattern on the top layer, an intermediate fabric layer and unpatterned conductive textile on the bottom layer (Supplementary Fig. 4). The phased surfaces play a crucial role in enabling the metamaterial structure to wirelessly interface with devices implanted in the body, serving to both capture signals emitted from an implant and focus surface waves propagating on the textile into the body on a target device. Previous studies on similar structures were conducted in the context of wireless power using copper strips excited by a coaxial cable[34]. We adapted the structure to support a feed structure in which the excitation is applied

from the side without reducing the magnitude of the excited currents (Fig. 2b, Supplementary Fig. 5), allowing it to be fabricated from planar conductive textiles. To focus the signal into the body, reactive elements (0.3–4.0 pF capacitors) control the phases of currents flowing through each ring, creating resonances within the 2.4–2.5 GHz industrial, scientific and medical band (ISM) frequency band (Supplementary Fig. 6). The phases of the currents flowing through the rings cover the entire $2\pi$ range required for complete wavefront control, as shown in Fig. 2e and Supplementary Fig. 6. The optimal phases are solved using the numerical scheme described in Supplementary Methods 1, based on the impedance matrix describing the mutual coupling between the rings and the target device. Owing to Lorentz reciprocity, the resulting structures can both efficiently receive radiation from implants and focus wireless signals to a target region greater, as illustrated at depths exceeding 4 cm in Supplementary Figs. 7 and 8.

The conformal propagation of the wireless signal along the body is facilitated by the SSP waveguide (Fig. 2c). To ensure that the fundamental surface mode supports frequencies within the desired ISM band, an analytical model of the surface plasmon mode dispersion (Fig. 2f) is utilized to design the structure's dimensions (Supplementary Methods 1)[40]. Following a previously established design procedure[40], we obtain a set of geometrical parameters that satisfies the requirements for surface wave propagation while having dimensions suitable for textile manufacturing. As shown in Fig. 2f, the structure parameter $h$ plays a key role in tuning the wavelength of the surface wave and the decay constant of the field above the surface. Simulations indicate that the surface currents of the fundamental mode flow along the comb edges and have a periodicity dictated by the spoof surface plasmon wavelength, rather than the spacing of the comb teeth (Supplementary Fig. 9).

The final component of the metamaterial design is the impedance matching section, which connects the SSP waveguide with phased surfaces at both ends (Fig. 2d). To make it compatible with textile fabrication processes and enable seamless integration into clothing, we developed a gradient matching section by gradually tapering a corrugated strip. The number of units $N$ is varied to minimize reflection from a 50 $\Omega$ port[43]. Figure 2g demonstrates that by selecting $N = 3$, we achieved a conversion loss of 0.94 dB.

The metamaterial textiles exhibit remarkable robustness to folding and bending, unlike conventional radio-frequency devices. Simulations demonstrate that the transmission loss is <2 dB for a U-turn with a radius of curvature of 1.25 mm (Supplementary Fig. 10). Furthermore, the SSP structure is capable of withstanding multidirectional bending and stretching, as demonstrated in more intricate deformations simulated in Supplementary Figs. 11, 12. Only when the phased surfaces are bent in the longitudinal direction with a radius of curvature <8 cm is significant signal degradation observed.

Networking of multiple wireless implants is also possible using a power divider structure with multiple phased surfaces. When a single implant terminal is excited, multiple phased current hotspots are generated, allowing for multinodal implant networking (Supplementary Figs. 13, 14). Although the wireless transmission in such a network is inversely proportional to the number of terminals, due to the averaged distribution of energy over multiple terminal phased surfaces, even with five implants at 4 cm depth, the transmission to each node is still about 34 dB higher than the case without metamaterial textiles, which is sufficient for reliable implant-to-implant communication.

In Fig. 2h, we demonstrate the integration of metamaterial textiles with a cotton polyester shirt. The entire metamaterial textile platform is fabricated by laser-cutting conductive textile (Cu/Ni polyester), which was then attached to the shirt with fabric adhesive. Full-wave simulations of implant communication in a human body model reveal that the textile surface is able to capture emitted signals, which propagate around the body curvature before being focused onto the implanted receiver, as demonstrated in Fig. 2i. In contrast, implant communication performed in the absence of the metamaterial textile is less efficient by more than three orders of magnitude due to the obstruction by the body and body-air mismatch. In addition, we evaluated the wireless communication of implants in different scenarios (Fig. 2j). The simulation results show that our metamaterial textiles can enhance the wireless communication by 30–40 dB, which translates into lower power consumption and higher data throughput of wireless communication[40].

## System characterization

We characterized the implant communication system by measuring wireless transmission in a water tank (50 cm × 45 cm × 30 cm, Supplementary Fig. 15). We investigated the dependence of the wireless networking performance on variations in the geometrical configuration, as schematically illustrated in Fig. 3a, b. Figure 3c, d shows that, although the wireless transmission generally decreases with increasing distance $L$ between the implant antennas or increasing implanted depth $z$, there is at least a 25 dB enhancement in all cases with the metamaterial textiles compared to that without. Hence, the textile platform can network implants that are deep in tissue ($z > 5$ cm) and distributed across the human body ($L > 35$ cm). The wireless performance is also robust to changes in orientation or alignment due to the movements of the implants or the phased surfaces. Figure 3h demonstrates that, despite the transverse polarization of the generated electric field relative to the phased surface, the signal can still be detected at all azimuthal angles $\theta$ without any nulls. Specifically, when the receiver is placed longitudinally ($\theta = 90°$), the transmission decreases to −82 dB, but still achieves over 10 dB enhancement compared to the radiative case without metamaterial textiles.

To assess the robustness of the signal enhancement, we conducted real-time monitoring of the wireless signal strength between two Bluetooth modules while pouring water into an acrylic container (see the "Methods" section and Supplementary Movie 1). The relative signal strength indicator (RSSI) between the two modules inside the container was recorded as the water was poured into the initially empty tank, followed by repeatedly attaching and removing the metamaterial textiles on its wall. As shown in Fig. 3i, a direct Bluetooth connection between the devices was maintained until the water level reached 1 cm above the device, at which point the RSSI dropped to −100 dB, indicating a disconnection. However, the wireless connection was immediately re-established when the metamaterial textile was placed on the wall of the tank. A comparison of RSSI shown in Fig. 3j shows the crucial role of metamaterial textiles in enabling wireless networking between the implants.

## Wireless networking and closed-loop control of implants

Our metamaterial textiles efficiently convert radiative waves to surface waves, facilitating robust radio wave propagation and enabling wireless networking of various implantable devices. This capability may open new opportunities for closed-loop bioelectronic therapies with networked devices equipped with sensing and actuating functions. For instance, diabetic patients could benefit from networked sensors and drug delivery implants for autonomous glucose level management[16] and paraplegics could recover motor function using a synchronized network of neural recorders and stimulators[44,45]. As a demonstration of networking capabilities, we successfully wirelessly networked two functional devices implanted in an adult living pig (45 kg) to achieve autonomous heart rate management.

We implanted a wireless loop recorder with two thin titanium electrodes in the right chest of an adult living pig (45 kg) and a VNS stimulator equipped with a commercial nerve cuff (3 mm inner diameter, 15 mm length, 1 mm electrode width, MicroProbes) in the neck. Each implant node was powered by a 100 mAh lithium-ion polymer battery and encapsulated with transparent silicone (see the "Methods" section, Supplementary Figs. 16–18). After the successful implantation of these devices, the metamaterial textile was placed over the body

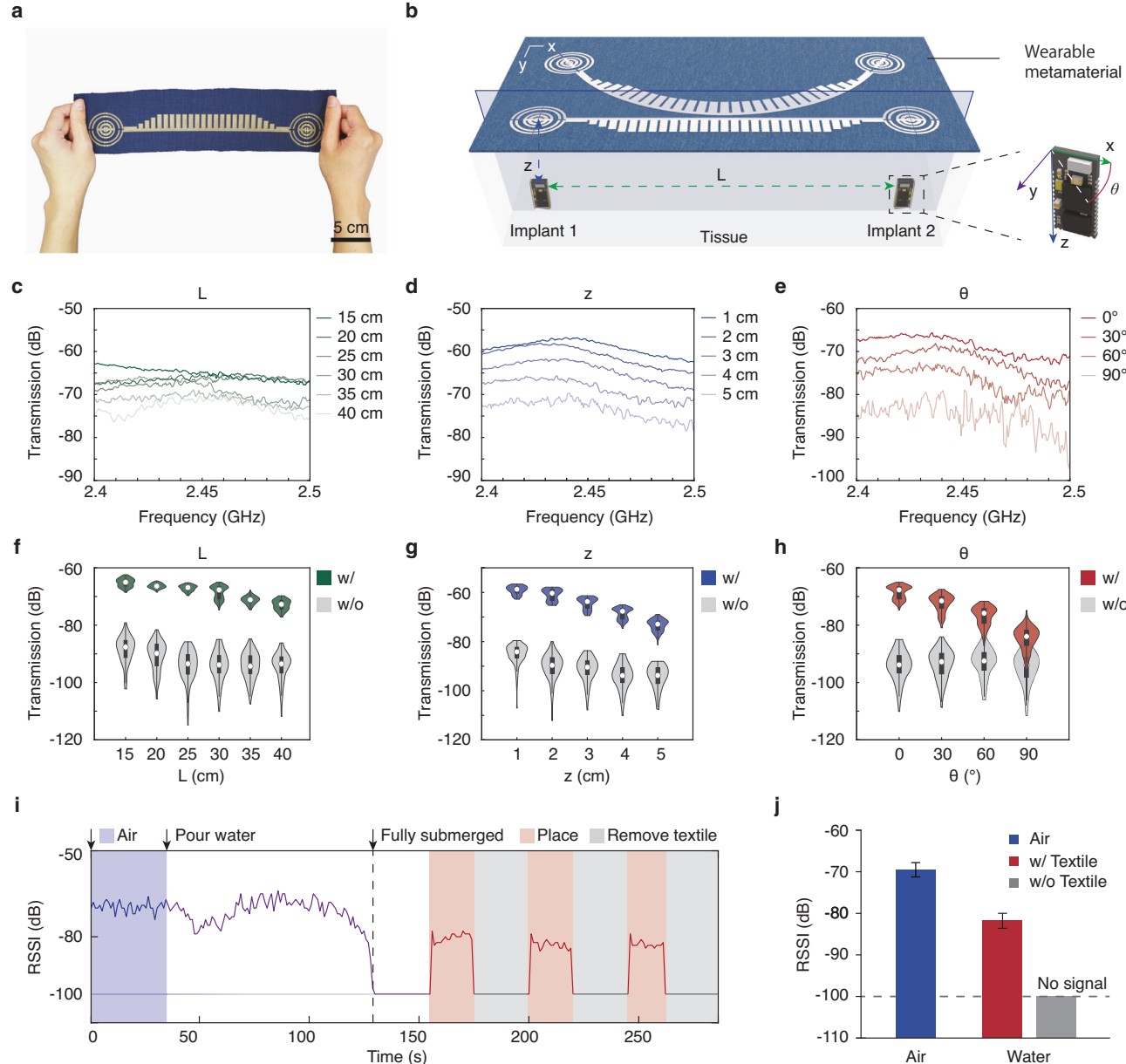

**Fig. 3 | Wireless communication performance of metamaterial textiles.**
**a** Photograph of the fabricated metamaterial textiles. Scale bar, 5 cm.
**b** Configuration of implant-to-implant wireless communication. Space below the metamaterial textile is assumed to be homogeneous muscle tissue. $L$ distance between two implant antennas, $z$ depth, $\theta$ azimuth angle. **c–e** Measured transmission spectra for different lengths of the metamaterial textile (**c**), depths of Bluetooth antennas in water (**d**), and rotation angles of the receiver antenna (**e**). **f–h** Violin plots for comparison of transmission coefficient measured in different configurations with and without the metamaterial textile at air–water interface over 2.4–2.5 GHz ISM band. Box plots inside the violins indicate the quartiles of corresponding transmission spectra. Endpoints show minimum and maximum values; white dots represent median values; whiskers denote 1.5 of the interquartile range. **i** Bluetooth RSSI recorded in Supplementary Movie 1. **j** Comparison of RSSI values. Error bars show mean ± s.d. of RSSI values across the indicated period in (**i**). Source data are provided as a Source Data file.

surface. Initial placement of the metamaterial textile relied on feedback from the implants without the use of imaging guidance (Supplementary Fig. 19). Once the textile placement was determined, wireless networking of the implants was repeatable and tolerant to misalignment (±1 cm, Supplementary Fig. 20). Computed tomography (CT) scans show the relative positions of the metamaterial textile and the implanted devices in a three-dimensional reconstruction (Fig. 4a and Supplementary Fig. 22a–c) and cross-sectional image (Fig. 4b and Supplementary Fig. 22d, e). The depth of implantation ranged from 2.5 to 3 cm from the skin surface.

Once networked, the loop recorder and VNS nodes establish a close-loop heart rate sensing and modulation system. Specifically,

the loop recorder continuously records the electrocardiogram (ECG) of the pig and calculates its heart rate. Once the heart rate exceeds a threshold $hr_t$, the loop recorder (ECG node) sends an instruction to the VNS node to start vagus nerve stimulation until the heart rate recovers below the threshold. The recorded ECG waveform and output stimulation signal from the VNS node are plotted in Fig. 4c, d. We also evaluated the wireless communication performance in the pig by measuring the RSSI and $|S_{21}|$ between the two nodes (Fig. 4e, f and Supplementary Fig. 23). Due to the long distance between these two implants, a direct conventional Bluetooth connection could not be established through tissues with RSSI below −100 dB. In contrast, our metamaterial textile brought

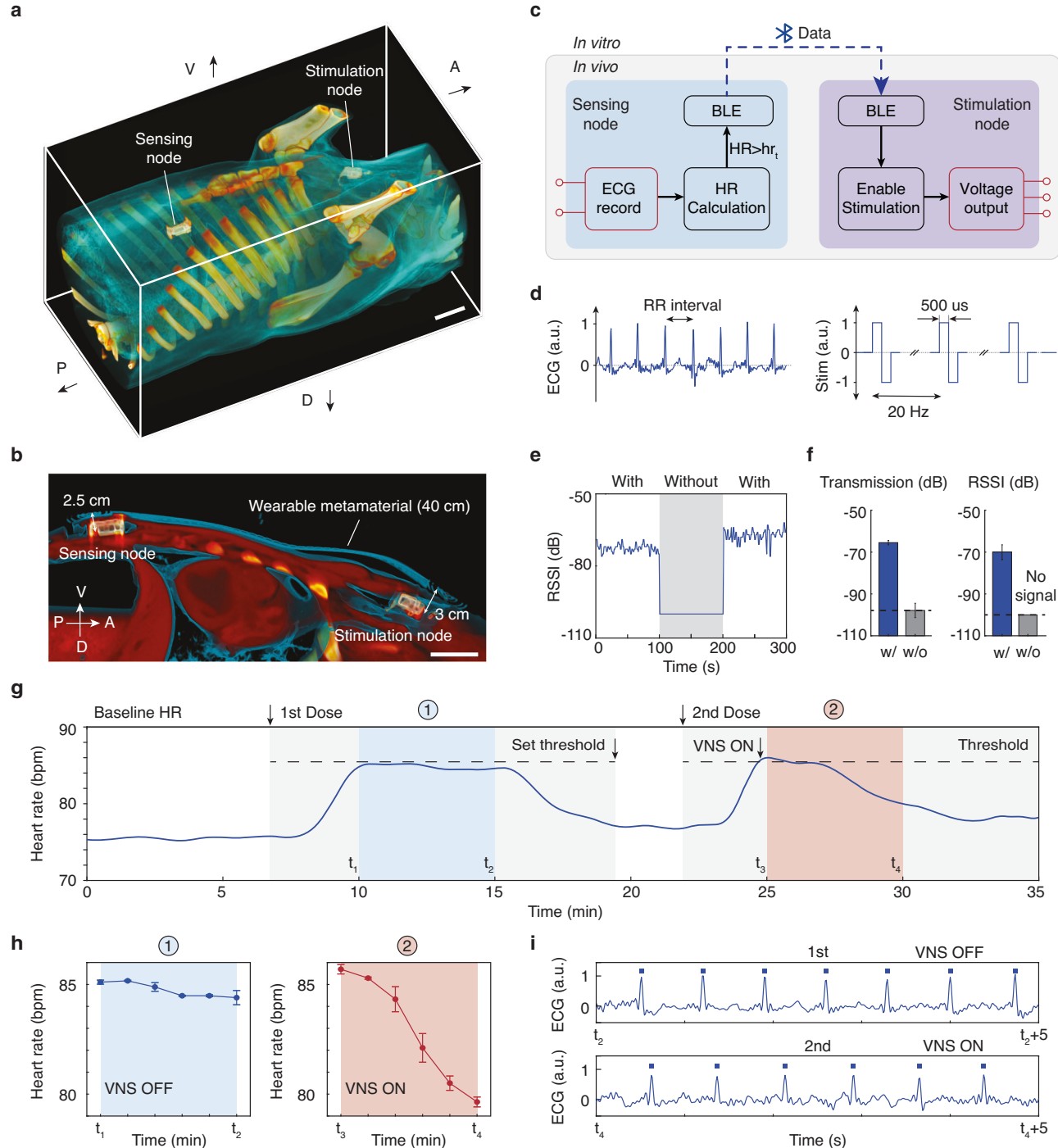

**Fig. 4 | In vivo wireless networking and closed-loop control of implantable devices in a pig. a** Three-dimensional computed tomography reconstruction of implants wireless networking in a pig model showing two devices (loop recorder and VNS node) implanted at the thorax and neck, respectively. Scale bar, 5 cm. **b** Computed tomography cross-section image showing the metamaterial textile, loop recorder, and VNS node. The distance between two implants and depths of implantation from the metamaterial textile are labeled. Scale bar, 5 cm. **c** Block diagram of the wireless closed-loop sensing (loop recorder) and stimulation (VNS node) system. When the detected heart rate (HR) increases above the threshold $hr_t$, VNS node will turn on the stimulation until the heart rate recovers. **d** Recorded ECG waveforms of the loop recorder with R-peak (RR) intervals labeled (left) and output stimulation signals of the VNS node (right). **e** RSSI received during Bluetooth interconnection of the two implanted nodes. **f** Comparison of $|S_{21}|$ (left) and RSSI (right) with and without the metamaterial textile. Error bars show mean ± s.d. of the transmission spectra within 2.4–2.5 GHz (left) and RSSI values in **e** (right). **g** Calculated heart rate during two dose injection cycles. Dashed black lines show the heart rate threshold $hr_t$. **h** 5-min change in heart rate after it plateaued during two trials as a function of time (error bars indicate mean ± s.d. of the previous 1-min heart rate). **i** Comparison of ECG signal segments 5 min after the heart rate plateaued. Fewer peaks detected in 5 s indicate lower heart rate. Peaks are marked with squares. Source data are provided as a Source Data file.

about a 33 dB enhancement, thus enabling an efficient wireless implant network.

We next illustrate in vivo sensing and therapy using our meta-material textile-enabled wireless implant network. Figure 4g shows the heart rate recorded from the loop recorder (Supplementary Figs. 24–26). After the first injection of phenylephrine (vasocon-strictor), we observed a rapid increase in heart rate, reaching a max-imum of 86 bpm, which was set as the stimulation threshold $hr_t$. Upon the second injection, the VNS node was automatically triggered once the heart rate exceeded the $hr_t$. Successful vagus nerve stimulation was verified through a comparison of 5-minute heart rate trends after plateauing during the two trials (Fig. 4h). In Fig. 4i, the ECG signal segment at $t_4$ shows a lower heart rate compared to $t_2$, resulting from the implant stimulation as seen from a reduction in detected peaks. Thus, the wireless networking capability was verified by in vivo demonstration of a closed-loop sensing-therapy implants system.

## Discussion

We have demonstrated a distributed, wireless network of bioelectronic implants that provides closed-loop control of physiological activity. Radio-frequency wireless connectivity between implants is established without an external relay using metamaterial textiles that passively amplify the received signal strength by more than three orders of magnitude. Experiments demonstrate the robustness of the network under a wide range of operating conditions, including displacement and creasing of the textile and rotation of the implants. In vivo, studies in a porcine model show the capability of the technology to wirelessly interlink distributed sensors and stimulators at the scale of the human body for adaptive heart rate control. Our approach is compatible with standard wireless protocols, such as Bluetooth Low Energy, that are clinically used by connected medical devices.

Future research efforts can focus on improving the orientation tolerance of the wireless network. Despite demonstrating a significant enhancement (>10 dB) even under the worst possible orientation, our current design relies on linearly polarized fields, rendering it sensitive to the relative orientation of the implant. To overcome this limitation, exploring metamaterials that support circularly polarized fields can enhance the robustness of wireless networking. Our present design facilitates low-power networking between devices implanted up to a depth of 5 cm, which is sufficient for a broad range of clinical applications[46]. By increasing the power of the emitted signal and optimizing the textile further, a greater depth of networking can be achieved. In particular, a larger diameter of the phase surface can significantly improve signal reception and transmission by increasing the aperture. While our textiles are robust to a wide range of defor-mations, excessive degrees of bending and twisting can degrade net-working performance. Further work may address this challenge by strategically placing more rigid textiles to limit deformation in the most sensitive regions of the metamaterial while preserving flexibility in regions essential for body motion. Additionally, future studies should address the possibility of networking more than two implanted devices, enabling more complex clinical applications such as the control of prosthetic limbs.

The clinical translation of our textile-based networking approach presents challenges in cost, manufacturing, reliability, and user adoption. However, our approach has several advantages that may facilitate adoption by users, including the fully passive nature of our textiles that simplifies manufacturing and enhances reliability. Addi-tionally, our textiles are designed to work with existing wirelessly enabled implants, which reduces regulatory hurdles. Furthermore, the rapid advances in conductive textiles in the industry can be harnessed to address challenges in cost, durability, and quality control. Ulti-mately, the success of our networking approach will depend on its ability to provide sufficient diagnostic or therapeutic benefits to justify the effort required to use it. Initial efforts may target applications in

which the textiles are worn temporarily during critical periods, such as monitoring during post-surgery recovery. As the technology matures, the technology may be used to target applications that require long-term use and/or have vital health functions, such as closed-loop drug delivery. Additionally, these wireless networking approaches may find applications beyond the domain of bioelectronic implants, enabling communication and sensing capabilities in various domains such as human–machine interfaces, ambient sensing, and automotive industries.

## Methods

### Research compliances

The animal experiment conformed to the Guide for the Care and Use of Laboratory Animals published by the National Institutes of Health, USA, and protocol approved by the Institutional Animal Care and Use Committee, National University of Singapore (R21-0377).

### Metamaterial textile design

The design of the metamaterial textile consists of an SSP waveguide, matching sections, and phased surface terminals. The SSP waveguide was designed by determining the geometrical parameters of the comb-like structure to support a surface mode with propagation constant $\beta$ approximating to the surface plasmon dispersion relation $\beta = \frac{\omega}{c}\sqrt{\varepsilon_1\varepsilon_2/(\varepsilon_1 + \varepsilon_2)}$, where $\varepsilon_1$ is the permittivity of the dielectric and $\varepsilon_2$ is the permittivity of the metal. The corresponding decay length in the upper region ($z > 0$) is given by $\alpha_1^{-1} = 1/\sqrt{\beta^2 - (\omega/c)^2}$. An unpatterned bottom layer was introduced to prevent the evanescent field from coupling into the lower body region. The intermediate fabric layer is characterized by thickness $t$ and permittivity $\varepsilon_{\text{textile}}$ based on wear-ability considerations. As $\beta$ is insensitive to $a$ and $b$, given the geo-metrical parameters of the structure ($a = 4$ mm, $b = 6$ mm, $p = 8$ mm, $t = 1.5$ mm) and $\varepsilon_{\text{textile}} = 1.5$, a set of dispersion curves of the SSP structure was obtained with varying $h$. The matching sections inter-facing the SSP waveguide and phased surfaces follow a gradient cor-rugated strip design. Transmission coefficients of the linearly tapered matching section were obtained by $|S_{21}|$ simulations with varying $N$ (number of matching units), yielding the value for efficient impedance conversion. The phased surface design comprises concentric split rings reactively loaded with passive elements. A numerical scheme based on a matched filter problem was used to determine the optimal currents (amplitudes and phases) and the corresponding passive ele-ments for focusing at a certain depth in muscle tissue (see Supple-mentary Methods 1).

### Numerical simulations

Electromagnetic simulations were carried out with CST Microwave Studio (Dassault Systems). To generate the optimal values of the pas-sive components, the scattering matrix was computationally obtained using a phased surface structure over homogeneous muscle tissue ($\varepsilon_r = 53.8 + i13.9$) with a receiver dipole (10 mm) placed at a depth of 4 cm. Dispersion curves of the SSP structure were obtained using the eigenmode solver for a unit cell defined with periodic boundary con-ditions. Field distributions were calculated using the finite-integration technique (FIT) using dipole excitation (at a depth of 4 cm) with and without the metamaterial structure placed above muscle tissue, using a computational voxel body model (Laura, CST Voxel Family) with a resolution of $1.875 \times 1.875 \times 1.25$ mm.

### Metamaterial textiles fabrication

A laser cutting machine (Universal Laser Systems, VLS 2.30) was used to fabricate conductive metamaterial textile patterns from adhesive Copper/Nickel polyester sheets (Conductive Non-woven Fabric 4770, Holland Shielding Systems). These patterns were attached to a cotton–polyester blend sweater for on-body system demonstration,

and to polyester fabric sheets for wireless communication experiments. Conductive epoxy (CW2460, Chemtronics) was used to bond passive components Supplementary Fig. 5) to the phased surface structure.

## Implant design and fabrication

The loop recorder (AD8232, Analog Devices) was configured to operate in two-electrode mode. A 0.5–40 Hz bandpass filter was implemented to obtain an ECG waveform with reduced motion artifacts. Two thin titanium electrodes (0.02 mm) were connected to the analog front-end (AFE) inputs. The VNS device was designed for constant biphasic current output with a dual-channel difference amplifier (AD8270, Analog Devices) fed by a boost-switching regulator (MIC2290YML-TR, Microchip Technology) and a switching voltage regulator (TC7660SCOA, Microchip Technology). A commercial nerve cuff (3 mm inner diameter, 15 mm length, 1 mm electrode width, MicroProbes) was assembled by soldering the exposed wires to the stimulation output of the VNS device. The loop recorder and VNS device were fabricated using a commercial printed circuit board process and then assembled with commercial ESP32 microcontroller modules (TinyPICO NANO), which were used to acquire and process the output signal from the loop recorder and wirelessly communicate the stimulation control parameters with VNS device via Bluetooth low-energy (BLE) protocols. Each implant node, connected to a 100 mAh lithium-ion polymer battery, was encapsulated with poly-dimethylsiloxane (PDMS, Dow Corning) while in deep sleep mode. A neodymium magnet (SparkFun Electronics) was used as an external wake-up source to actuate the reed switch (CT05, COTO Technology) embedded within both nodes before in vivo experiments.

## Bench-top evaluation

Benchtop evaluations were performed by placing the metamaterial textile on the wall of a rectangular acrylic container (width 45 cm, height 30 cm, length 50 cm, and wall thickness 5 mm) filled with water. The transmission was measured as $|S21|$ between two identical antennas (NN01-102 Antenna Evaluation Board, Ignion) connected to a vector network analyzer (N9915A, Keysight) using coaxial cables (SMA-SMA, 50 Ω, Amphenol). Wireless communication was performed using two implant nodes through Bluetooth links, during which changes in received signal strength were recorded. Controls were performed by repeating the experiments with an unpatterned textile.

## Large animal experiment

Experiments used a pig (Species: Sus scrofa domestica; Strain: Yorkshire × Landrace; Sex: female; Age: 6 months; Weight: 45 kg) model ($n = 1$) supplied by Singapore's National Large Animals Research Facility to demonstrate wireless communication between implants with metamaterial textiles. Sex was not considered in the study design, because it does not affect the wireless networking demonstration. The pig was pre-medicated with intramuscular ketamine (10 mg/kg, Ceva), midazolam (0.6 mg/kg, Cheplapharm), and atropine (0.04 mg/kg, Ilium), induced with 4% isoflurane (Baxter), intubated and maintained under anesthesia with 1–2% isoflurane throughout the experiments. Once stable anesthesia was obtained, an incision was made in the skeletal muscle of the thoracic region with the muscles exposed by blunt dissection. The loop recorder (30 mm length, 15 mm width, 10 mm thickness) was placed in the muscle region. On the right side of the neck, another incision was made to implant the VNS node (30 mm length, 15 mm width, 10 mm thickness) with the nerve cuff attached to the right cervical vagus nerve (Supplementary Fig. 21). The corresponding skin incisions were closed using surgical suturing, and the metamaterial textile was placed over the skin, during which the position was adjusted accordingly. The comparison of wireless communications efficiency ($|S_{21}|$ and RSSI measurement) between the two implants was then performed. Next, phenylephrine (vasoconstrictor, Sandoz) was administered via intravenous (IV) injection (0.44 μg/kg) to raise the cardiac output. Heart rate was continuously monitored by the loop recorder. After the second dose injection, the VNS node, wirelessly controlled by the loop recorder, generated biphasic pulses (20 Hz and 500 μs pulse width) across the cuff electrodes. The animal was euthanized upon completion of the experiment. CT scan was performed immediately post-mortem.

## Statistics and reproducibility

No statistical method was used to predetermine the sample size. No data were excluded from the analyses. The experiments were not randomized. The Investigators were not blinded to allocation during experiments and outcome assessment.

## Reporting summary

Further information on research design is available in the Nature Portfolio Reporting Summary linked to this article.

## Data availability

All data supporting the findings of this study are available within the article and its supplementary files. Any additional requests for information can be directed to, and will be fulfilled by, the lead contact. Source data are provided with this paper.

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

## Acknowledgements

J.S.H. acknowledges support from the National Research Foundation Singapore (NRF2021-NRF-ANR004) and the Advanced Research and Technology Innovation Centre (HFM-RP6). B.C.K.T. acknowledges support of RIE2020 Advanced Manufacturing and Engineering Programmatic Grant A18A1b0045 and Q.Z. acknowledges support from the National University of Singapore Research Scholarship.

## Author contributions

X.T., Q.Z., and J.S.H. conceived and planned the research. X.T. and Q.Z. designed the metamaterial textiles, conducted the theoretical studies and simulations, and characterized the wireless networking system. X.T., Q.Z., S.A.K., R.R.L., C.J.C., and J.S.H. performed the in vivo experiments. D.T.N., Z.X., Z.L., X.Y., and X.X. supported system characterization. C.W., B.C.K.T., D.N., and C.J.C. contributed to the study design. X.T., Q.Z., and J.S.H. wrote the paper with input from all the authors.

## Competing interests

The authors declare no competing interests.
