## [Peer review file · Nature Communications]

REVIEWER COMMENTS

Reviewer #1 (Remarks to the Author):

While the article presents a promising development in the field of wireless networking of bioelectronic implants, there are some potential areas for improvement:

1) The introduction is clear and all the objectives well stated. At the same time, some recent technology developments are missing such as:

- _ metastructures [Inverse-designed metastructures that solve equations, *Science* 363 (6433), 1333-1338, 2019]
- _ nanoparticles [Targeted dielectric coating of silver nanoparticles with silica to manipulate optical properties for metasurface applications, *Materials Chemistry and Physics*, 126250, 2022]
- _ near-zero-index materials [On the performance of an ENZ-based sensor using transmission line theory and effective medium approach, *New Journal of Physics* 21 (4), 043056, 2019]
- _ graphene [The graphene field effect transistor modeling based on an optimized ambipolar virtual source model for DNA detection, *Applied Sciences* 11 (17), 8114, 2021]
- _ plasmonics [Plasmonic Optical and Chiroptical Response of Self-Assembled Au Nanorod Equilateral Trimers, *ACS nano*, 2019]
- _ multi-functional structures [Multifunctional composites: A metamaterial perspective. *Multifunctional Materials*, 2(4), 043001, 2019]

It would be beneficial for the reader if authors include such technologies in the introduction section to have a complete picture of the state-of-art.

2) An analytical model to describe the structure behavior is missing. Consider the following mathematical approaches. Can they be applied to your paper? If yes why? If not, justify as well.

- _ Scattering/Polarizability [Electromagnetic and thermal nanostructures: from waves to circuits, *Engineering Research Express* 2 (1), 015045, 2020]
- _ Green's function [Spectral dyadic Green's function formulation for planar integrated structures, *IEEE Transactions on Antennas and Propagation* 36 , 8, 1988]
- _ Field and Circuit Theory [Pozar, David M. "Microwave engineering education: From field theory to circuit theory." 2012 IEEE/MTT-S International Microwave Symposium Digest. IEEE, 2012].

Explain what are the main differences of your model with this one above mentioned

3) To explore the device behaviour, authors can consider the following interesting electromagnetic phenomena: electric/magnetic surface currents, and displacements currents.

Include such phenomena in your model and explain how they can affect the device properties.

4) The paper lack in application examples. Take into consideration the following: sensing and diagnostics, biomedicine, telecommunications, absorbers, measurements, nanoelectronics, automotive.

I would suggest to create a small paragraph by considering such applications and explaining how you can use your device for them.

Please highlight what's new in yours.

5) No limitations of the proposed method have been highlighted.

6) No future improvements/works have been discussed.

7) While the metamaterial textile used in this study demonstrated impressive amplification of the received wireless signal, there are still potential areas for improvement. One limitation of the current design is that it requires a specific orientation to achieve the optimal wireless communication performance. Future research could explore the development of metamaterials that are more tolerant to variations in orientation or alignment.

8) Additionally, while the metamaterial textile was effective in amplifying wireless signals for implants at depths of up to 5 cm, further optimization may be required for implants at even greater depths. This could involve exploring different geometries and materials for the metamaterial.

9) Finally, the current study focused on wireless communication between two implants. In future work, the metamaterial textile could be tested in more complex scenarios, such as networks of multiple implants, to determine its scalability and potential for widespread use in clinical settings.

Reviewer #2 (Remarks to the Author):

Tian and coauthors presented a technology for wireless networking between implantable devices. The technology was realized by radiative wave – to – surface wave conversion and propagation of radio surface plasmons by metamaterial textiles. The system design was carefully introduced throughout the report, and its performance was verified quantitatively by comparing to that of the system without the metamaterial. The system was also verified by in vivo demonstrations showing a closed-loop control of the heartbeat of an alive pig. Using the unique system, the authors successfully suggested an interesting way to build a network between bioelectronic devices. This work is a meaningful contribution to the field of soft bioelectronics. In light of the demand, novelty, and quality of this work, the reviewer recommends publication of this manuscript in Nature communications after minor revision. To improve the quality of the current manuscript, following recommendations can be made.

Comment #1: The authors compared the performance of the network system with and without the metamaterial textiles throughout the report. The report could be improved if the quantitative and qualitative comparison between the wireless relay systems are added. One of the key advantages of the proposed system is that the signal can be propagated passively via metamaterial textiles. As a result, the energy efficiency of the overall implant-to-implant wireless communication system can be increased compared to the conventional communication system. The reviewer recommends authors compare the energy efficiency of the proposed system and the conventional systems quantitatively and add related texts for discussion.

Comment #2: The proposed system was demonstrated by using a porcine model, which confirmed that the communication distance can reach up to 40 cm. However, it seems that the system characterization could not demonstrate the performance at such distance. It would be better to revise related data in a wider range.

Comment #3: The manuscript claims that the network is formed on a textile by verifying the robust performance under folding deformations and by fabricating the device with passive elements only. In some sense, the construction conforms to the characteristic of a textile. However, a textile generally undergoes more complex deformations (e.g. crumpling, stretching) than the unidirectional folding. It would be better to add additional components that impose limitations on the folding direction and stretch to minimize signal degradation.

Comment #4: Although soft devices based on textiles are very attractive, they have not been commercialized yet (at least in a mass scale). There can be many issues, and reliable mass production and performance/quality guarantee for long-term customer use would be challenging issues. More discussions in terms of technology transfer to the industry can be added for readers.

Reviewer #3 (Remarks to the Author):

In this work, the authors reported a direct implant-to-implant wireless networking at the scale of the human body using metamaterial textiles. The textiles facilitate non-radiative propagation of radio-frequency signals along the surface of the body. The authors' group has reported the wireless networking at the scale of the human body using metamaterial textiles (Nature Communications, 2020, 11, 444; Nature Communications, 2022, 13, 2190; Nature Electronics, 2021, 4, 382–39) And the implantable sensor network was also reported in Nature Electronics, 2019, 2, 335–342). All these previous publications largely undermined the novelty of current manuscript. I do not see enough advancement in current work to justify a publication in Nature Communications.

Re: Nature Communications manuscript NCOMMS-23-05814-T, Tian, et al “Implant-to-implant wireless networking with metamaterial textiles”

To the reviewers,

We thank all the reviewers for their insightful and constructive comments on our manuscript. Below, please find a point-to-point response to the comments. As detailed below, we have extensively revised the main text to address the concerns raised, which are highlighted in blue in the revised manuscript. We believe that the manuscript is much improved as a result.

Reviewer #1

“While the article presents a promising development in the field of wireless networking of bioelectronic implants, there are some potential areas for improvement:”

Reply: We thank the reviewer for the positive view of the manuscript.

1. The introduction is clear and all the objectives well stated. At the same time, some recent technology developments are missing such as:

_ metastructures [Inverse-designed metastructures that solve equations, Science 363 (6433), 1333-1338, 2019]

_ nanoparticles [Targeted dielectric coating of silver nanoparticles with silica to manipulate optical properties for metasurface applications, Materials Chemistry and Physics, 126250, 2022]

_ near-zero-index materials [On the performance of an ENZ-based sensor using transmission line theory and effective medium approach, New Journal of Physics 21 (4), 043056, 2019]

_ graphene [The graphene field effect transistor modeling based on an optimized ambipolar virtual source model for DNA detection, Applied Sciences 11 (17), 8114, 2021]

_ plasmonics [Plasmonic Optical and Chiroptical Response of Self-Assembled Au Nanorod Equilateral Trimers, ACS nano, 2019]

_ multi-functional structures [Multifunctional composites: A metamaterial perspective. Multifunctional Materials, 2(4), 043001, 2019]

It would be beneficial for the reader if authors include such technologies in the introduction section to have a complete picture of the state-of-art.

Reply: We thank the reviewer for the suggestions to improve the introduction section of the manuscript. As requested, we have revised the main text to reference more recent advances in metamaterials, include several of the suggested references. Because the scope of our manuscript is limited to radio-frequency communication, references related to nano-optics and two-dimensional materials were not included.

“Here, we demonstrate an approach to achieve direct implant-to-implant wireless networking of bioelectronic implants across the human body (Fig. 1a). Our approach employs wearable metamaterials – artificial materials with subwavelength structure that enable extraordinary control over optical, acoustic, and radio-frequency fields^{27,28}. Metamaterials have found broad applications in wireless technology, ranging from wireless communication^{29,30} to remote sensing³¹⁻³³, wireless power transfer³⁴⁻³⁶, and wave-based computing^{37,38}. To implement our approach, we design metamaterial textiles that can passively facilitate the non-radiative propagation of radio-frequency signals along the surface of the body³⁹⁻⁴¹, enabling direct communication between implants using established wireless protocols (Fig. 1b–c). We establish wireless connectivity between the textile and implants by coherently amplifying the signal on the surface of the body using phased textile structures (Fig. 1d). These metamaterial textiles can be easily integrated into regular clothing⁴² and are compatible with the Bluetooth Low Energy standard used in commercially available medical devices.”

2. An analytical model to describe the structure behavior is missing. Consider the following mathematical approaches. Can they be applied to your paper? If yes why? If not, justify as well.

_ Scattering/Polarizability [Electromagnetic and thermal nanostructures: from waves to circuits, Engineering Research Express 2 (1), 015045, 2020]

_ Green's function [Spectral dyadic Green's function formulation for planar integrated structures, IEEE Transactions on Antennas and Propagation 36 , 8, 1988]

_ Field and Circuit Theory [Pozar, David M. "Microwave engineering education: From field theory to circuit theory." 2012 IEEE/MTT-S International Microwave Symposium Digest. IEEE, 2012].

Explain what are the main differences of your model with this one above mentioned.

Reply: We thank the reviewer for raising this point regarding an analytical model. We would like to point out that a theoretical analysis of phased surface is included in the supplementary methods (Supplementary Methods 1a), which can be directly used for design. Regarding the electromagnetic behavior of the spoof surface plasmon structure, our work makes use of existing analytical models^[R1,2] based on Floquet mode analysis. This is sufficient to describe the key aspects of the design without resorting to Green's functions or the circuit theory of fields. We agree that including an explicit analysis for our design can be helpful for readers, although we

emphasize that it is based on previously published work. We have added the analysis in Supplementary Methods 1b and also revised the Section III.

“The metamaterial textile structure comprises a spoof surface plasmonic (SSP) waveguide⁴⁰, terminated by impedance matching sections⁴³ and phased surface structures³⁴ at both ends, as illustrated in Fig. 2a.”

“The optimal phases are solved using the numerical scheme described in Supplementary Methods 1, based on the impedance matrix describing the mutual coupling between the rings and the target device. Owing to Lorentz reciprocity, the resulting structures can both efficiently receive radiation from implants and focus wireless signals to a target region greater, as illustrated at depths exceeding 4 cm in Supplementary Fig. 7,8.”

“The conformal propagation of the wireless signal along the body is facilitated by the SSP waveguide (Fig. 2c). To ensure that the fundamental surface mode supports frequencies within the desired ISM band, an analytical model of the surface plasmon mode dispersion (Fig. 2f) is utilized to design the structure's dimensions (Supplementary Methods 1)⁴⁰.”

[R1] Shen, X., Cui, T. J., Martin-Cano, D. & Garcia-Vidal, F. J. Conformal surface plasmons propagating on ultrathin and flexible films. *Proc. Natl Acad. Sci. USA* **110**, 40–45 (2013).

[R2] Tian, X. et al. Wireless body sensor networks based on metamaterial textiles. *Nat. Electron.* **2**, 243–251 (2019).

3. To explore the device behaviour, authors can consider the following interesting electromagnetic phenomena: electric/magnetic surface currents, and displacements currents. Include such phenomena in your model and explain how they can affect the device properties.

Reply: We thank the reviewer for raising this interesting point. We agree that, similar to other metallic structures at radio-frequencies, surface currents can provide an alternative approach to study the behavior of electromagnetic devices. In addition to the surface currents on the phased surface shown in Supplementary Fig. 6, we have included an additional supplementary figure (Supplementary Fig. 9) showing the surface currents on the SSP structure. Current density analysis of the phased surface shows that the coupling of resonances through the mutual impedances of the rings yields phases capable of covering the entire 2π range. These surface currents are essential the behavior of the device because the enable rapid phase changes across the deeply subwavelength spacing between the rings.

We have revised Section III to describe the behavior of the surface currents.

“Previous studies on similar structures were conducted in the context of wireless power using copper strips excited by a coaxial cable³⁴. We adapted the structure to support a feed structure in which the excitation is applied from the side without reducing the magnitude of the excited currents (Fig. 2b, Supplementary Fig. 5), allowing it to be fabricated from planar conductive textiles. To

focus the signal into the body, reactive elements (0.3–4.0 pF capacitors) control the phases of currents flowing through each ring, creating resonances within the 2.4–2.5 GHz industrial, scientific and medical band (ISM) frequency band (Supplementary Fig. 6). The phases of the currents flowing through the rings cover the entire 2π range required for complete wavefront control, as shown in Fig. 2e and Supplementary Fig. 6.”

“Following a previously established design procedure⁴⁰, we obtain a set of geometrical parameters that satisfies the requirements for surface wave propagation while having dimensions suitable for textile manufacturing. As shown in Fig. 2f, the structure parameter h plays a key role in tuning the wavelength of the surface wave and the decay constant of the field above the surface. Simulations indicate that the surface currents of the fundamental mode flow along the comb edges and have a periodicity dictated by the spoof surface plasmon wavelength, rather than the spacing of the comb teeth (Supplementary Fig. 9).”

Supplementary Figure 9. Surface current of the SSP waveguide. The simulated surface current vector distributions of the spoof surface plasmonic mode.

4. The paper lack in application examples. Take into consideration the following: sensing and diagnostics, biomedicine, telecommunications, absorbers, measurements, nanoelectronics, automotive. I would suggest to create a small paragraph by considering such applications and explaining how you can use your device for them. Please highlight what's new in yours.

Reply: We thank the reviewer for the suggestion to include a paragraph discussing applications. We have added a paragraph with application examples within the biomedical domain in the Discussion section and also briefly highlight applications relevant to our porcine model study in Section V.

“Ultimately, the success of our networking approach will depend on its ability to provide sufficient diagnostic or therapeutic benefits to justify the effort required to use it. Initial efforts may target applications in which the textiles are worn temporarily during critical periods, such as monitoring during post-surgery recovery. As the technology matures, the technology may be used to target applications that require long-term use and/or have vital health functions, such as closed-loop

drug delivery. Additionally, these wireless networking approaches may find applications beyond the domain of bioelectronic implants, enabling communication and sensing capabilities in various domains such as human-machine interfaces, ambient sensing, and automotive industries.”

“Our metamaterial textiles efficiently convert radiative waves to surface waves, facilitating robust radio wave propagation and enabling wireless networking of various implantable devices. This capability may open new opportunities for closed-loop bioelectronic therapies with networked devices equipped with sensing and actuating functions. For instance, diabetic patients could benefit from networked sensors and drug delivery implants for autonomous glucose level management¹⁶ and paraplegics could recover motor function using a synchronized network of neural recorders and stimulators^{44,45}. As a demonstration of networking capabilities, we successfully wirelessly networked two functional devices implanted in an adult living pig (45 kg) to achieve autonomous heart rate management.”

5. *No limitations of the proposed method have been highlighted.*

Reply: We thank the reviewer for raising this important point. We have added a new paragraph in the Discussion section to clarify the limitations of proposed method.

“Future research efforts can focus on improving the orientation tolerance of the wireless network. Despite demonstrating a significant enhancement (>10 dB) even under the worst possible orientation, our current design relies on linearly polarized fields, rendering it sensitive to the relative orientation of the implant. To overcome this limitation, exploring metamaterials that support circularly polarized fields can enhance the robustness of wireless networking. Our present design facilitates low-power networking between devices implanted up to a depth of 5 cm, which is sufficient for a broad range of clinical applications⁴⁶. By increasing the power of the emitted signal and optimizing the textile further, greater depth of networking can be achieved. In particular, a larger diameter of the phase surface can significantly improve the signal reception and transmission by increasing the aperture. While our textiles are robust to a wide range of deformations, excessive degrees of bending and twisting can degrade networking performance. Further work may address this challenge by strategically placing more rigid textiles to limit deformation in the most sensitive regions of the metamaterial while preserving flexibility in regions essential for body motion. Additionally, future studies should address the possibility of networking more than two implanted devices, enabling more complex clinical applications such as the control of prosthetic limbs.”

6. *No future improvements/works have been discussed.*

Reply: We thank the reviewer for raising this important point. Same as the response to comments #5, we have added two new paragraphs in the Discussion section to clarify the limitations of proposed method and related future works.

We also have included a new paragraph in the Discussion section about future commercialization.

“The clinical translation of our textile-based networking approach presents challenges in cost, manufacturing, reliability, and user adoption. However, our approach has several advantages that may facilitate adoption by users, including the fully passive nature of our textiles that simplifies manufacturing and enhances reliability. Additionally, our textiles are designed to work with existing wirelessly-enabled implants, which reduces regulatory hurdles. Furthermore, the rapid advances in conductive textiles in the industry can be harnessed to address challenges in cost, durability, and quality control. Ultimately, the success of our networking approach will depend on its ability to provide sufficient diagnostic or therapeutic benefits to justify the effort required to use it. Initial efforts may target applications in which the textiles are worn temporarily during critical periods, such as monitoring during post-surgery recovery. As the technology matures, the technology may be used to target applications that require long-term use and/or have vital health functions, such as closed-loop drug delivery. Additionally, these wireless networking approaches may find applications beyond the domain of bioelectronic implants, enabling communication and sensing capabilities in various domains such as human-machine interfaces, ambient sensing, and automotive industries.”

7. While the metamaterial textile used in this study demonstrated impressive amplification of the received wireless signal, there are still potential areas for improvement. One limitation of the current design is that it requires a specific orientation to achieve the optimal wireless communication performance. Future research could explore the development of metamaterials that are more tolerant to variations in orientation or alignment.

Reply: We thank the reviewer for noting the advancement of our work and raising the important challenge of orientation. Alignment between the implanted antenna and the metamaterial is indeed required for optimal wireless communication performance. However, orientation also is a challenge that exists for any wireless communication system, regardless of whether our metamaterial textiles are utilized. The degree to which the performance depends on orientation follows the same $\sin \theta$ dependency as a conventional pair of linearly polarized antennas, with the slight modification that θ is the angle between the polarization of the implant antenna and the phase surface. We note that even in the worst orientation, the metamaterial offers > 10 dB enhancement in the transmission efficiency.

We fully agree that the tolerance for orientation could be improved in future research. In particular, the design of metamaterial structures that support circularly polarized radiative modes could eliminate sensitivity to orientation in the transverse plane. We have included a discussion on future work in the main manuscript:

“Future research efforts can focus on improving the orientation tolerance of the wireless network. Despite demonstrating a significant enhancement (>10 dB) even under the worst possible orientation, our current design relies on linearly polarized fields, rendering it sensitive to the

relative orientation of the implant. To overcome this limitation, exploring metamaterials that support circularly polarized fields can enhance the robustness of wireless networking.”

8. Additionally, while the metamaterial textile was effective in amplifying wireless signals for implants at depths of up to 5 cm, further optimization may be required for implants at even greater depths. This could involve exploring different geometries and materials for the metamaterial.

Reply: We thank the reviewer for this insightful point. Indeed, our present manuscript demonstrates signal amplification at depths up to 5 cm, which covers many clinical applications of interest for implant networks (e.g. peripheral nerve stimulators and subcutaneously implanted sensors)^[R3]. We fully agree, however, that some specialized applications, such as those involving gastrointestinal implants and/or targeting overweight patient populations, may require greater depths of operation^[R4,5].

- Greater operating depth can be most straightforwardly achieved by increasing the transmitted power. We have included simulations that show that our metamaterial textiles still significantly enhance wireless signals and network implants at depths up to 8 cm (Supplementary Fig. 15). In our present experiments, the transmit power of the BLE module is set to only 0 dBm (1 mW) to maximize the battery life of the implants. This power level can be increased up to 20 dBm (100 mW) to extend the operating depth while complying with SAR limits.
- As noted by the reviewer, further optimization of the metamaterial may improve the operating depth when increasing the transmit power becomes impractical due to power or safety constraints. In particular, increasing the diameter of the phase surface could lead to marked improvements due to the increase aperture for both receiving signals and focusing the wireless signal^[R6].

To clarify this point, we have included a discussion for implant depths in the new Discussion section.

“Our present design facilitates low-power networking between devices implanted up to a depth of 5 cm, which is sufficient for a broad range of clinical applications⁴⁶. By increasing the power of the emitted signal and optimizing the textile further, greater depth of networking can be achieved. In particular, a larger diameter of the phase surface can significantly improve the signal reception and transmission by increasing the aperture.”

[R3] Ho, J. S. et al. Planar immersion lens with metasurfaces. *Phys. Rev. B* **91**, 125145–125148 (2015).

[R4] A. Sani, A. Alomainy and Y. Hao. Numerical Characterization and Link Budget Evaluation of Wireless Implants Considering Different Digital Human Phantoms. *IEEE Trans. Microwave Theory Tech.* **57**, 2605-2613 (2009).

[R5] J. Abouei, J. D. Brown, K. N. Plataniotis and S. Pasupathy. Energy Efficiency and Reliability in Wireless Biomedical Implant Systems. *Trans. Inf. Technol. Biomed.* **15**, 456-466 (2011).

[R6] Ho, J. S. et al. Wireless power transfer to deep-tissue microimplants. *Proc. Natl Acad. Sci. USA* **111**, 7974–7979 (2014).

9. Finally, the current study focused on wireless communication between two implants. In future work, the metamaterial textile could be tested in more complex scenarios, such as networks of multiple implants, to determine its scalability and potential for widespread use in clinical settings.

Reply: We thank the reviewer for the insightful suggestion. We fully agree that our metamaterial textile could be applied and tested in more complex scenarios in future work. Our simulations demonstrating the capability of the metamaterial textile to support multi-implant networking capabilities may provide a starting point for such investigations (Supplementary Fig. 9 and 10). We have included a discussion of the use of our metamaterials for multiple implants in the revised manuscript:

“Additionally, future studies should address the possibility of networking more than two implanted devices, enabling more complex clinical applications such as the control of prosthetic limbs.”

Reviewer #2

“Tian and coauthors presented a technology for wireless networking between implantable devices. The technology was realized by radiative wave – to – surface wave conversion and propagation of radio surface plasmons by metamaterial textiles. The system design was carefully introduced throughout the report, and its performance was verified quantitatively by comparing to that of the system without the metamaterial. The system was also verified by in vivo demonstrations showing a closed-loop control of the heartbeat of an alive pig. Using the unique system, the authors successfully suggested an interesting way to build a network between bioelectronic devices. This work is a meaningful contribution to the field of soft bioelectronics. In light of the demand, novelty, and quality of this work, the reviewer recommends publication of this manuscript in Nature communications after minor revision. To improve the quality of the current manuscript, following recommendations can be made.”

Reply: We thank the reviewer for noting the novelty, quality and general interest of our work.

1. The authors compared the performance of the network system with and without the metamaterial textiles throughout the report. The report could be improved if the quantitative and qualitative comparison between the wireless relay systems are added. One of the key advantages of the proposed system is that the signal can be propagated passively via metamaterial textiles. As a result, the energy efficiency of the overall implant-to-implant wireless communication system can be increased compared to the conventional communication system. The reviewer recommends authors compare the energy efficiency of the proposed system and the conventional systems quantitatively and add related texts for discussion.

Reply: We thank the reviewer for raising the comparison with conventional wireless relay systems. We have included a new Fig. 1f and Supplementary Fig. 3 to illustrate the comparison with the relay. We have also revised the section System Overview to describe this comparison.

“... Full-wave electromagnetic simulations show that the implant-to-implant transmission efficiency is over three orders of magnitude (>30 dB) higher than without the metamaterial textile for distances ranging from 10–50 cm (Fig. 1f). A comparison with a relay system that uses radiative communication devices placed directly over the sensor and stimulator shows an improvement in transmission efficiency of more than 20 dB (Fig. 1f). The subwavelength structuring of the metamaterial plays a crucial role in achieving this enhanced transmission, as unstructured conductive textiles yield transmission efficiencies comparable to no textiles (Supplementary Fig. 1–3).”

Supplementary Figure 3. Comparison of the transmission coefficient as a function of distance. **a**, Illustrations of different wireless implant-to-implant communication methods. **b**, received power as a function of distance. The devices are implanted in varied depth (z), from 3 to 5cm.

2. The proposed system was demonstrated by using a porcine model, which confirmed that the communication distance can reach up to 40 cm. However, it seems that the system characterization could not demonstrate the performance at such distance. It would be better to revise related data in a wider range.

Reply: We thank the reviewer for the careful review of the manuscript. We have added a new measurement ($L=40$ cm) and revised the two main figure panels (Fig. 3c, f) to address this point.

Figure 3. Wireless communication performance of metamaterial textiles. c, Measured transmission spectra for different lengths of the metamaterial textile. **f,** Violin plots for comparison of transmission coefficient measured with and without the metamaterial textile at air-water interface over 2.4–2.5 GHz ISM band.

3. The manuscript claims that the network is formed on a textile by verifying the robust performance under folding deformations and by fabricating the device with passive elements only. In some sense, the construction conforms to the characteristic of a textile. However, a textile generally undergoes more complex deformations (e.g. crumpling, stretching) than the unidirectional folding. It would be better to add additional components that impose limitations on the folding direction and stretch to minimize signal degradation.

Reply: We thank the reviewer for raising this point. Although we acknowledge that textiles can undergo more complex deformations (e.g., crumpling and stretching) than unidirectional folding, we would like to point out that our design can withstand a certain degree of deformation. To illustrate this point. We have included new supplementary figures (Supplementary Fig. 11, 12), which depict multidirectional bending simulations. The simulations show that SSP structure is robust to multidirectional bending and stretching, and obvious signal degradation only could be observed when phased surfaces are bended in the x-axis with a radius of curvature less than 8 cm. In addition, the thickness of the conductive fabrics used here (0.3 mm) also imposes some limitations on the degree of deformation and can prevent excessive bending to some degree.

We agree that beyond a certain degree, the metamaterial textiles may not function effectively, necessitating the greater limitations on degree of deformation. To achieve this, we could

selectively make parts of the conductive textiles more rigid in strategic regions around the body to limit the degree of deformation that the entire structure undergoes. The key is to balance the rigidity of different areas: the phased surface is the most sensitive component and should be the most rigid, while the SSP is less sensitive. In this way, future metamaterial textiles could incorporate rigid islands interconnected by the textiles in areas where rigidity is critical, such as the chest. In contrast, areas that require flexibility, such as sleeves, can utilize more flexible textiles.

We have added a new sentence in System Design section and a new paragraph in Discussion section.

“The metamaterial textiles exhibit remarkable robustness to folding and bending, unlike conventional radio-frequency devices. Simulations demonstrate that the transmission loss is less than 2 dB for a U-turn with a radius of curvature of 1.25 mm (Supplementary Fig. 10). Furthermore, the SSP structure is capable of withstanding multidirectional bending and stretching, as demonstrated in more intricate deformations simulated in Supplementary Fig. 11–12. Only when the phased surfaces are bent in the longitudinal direction with a radius of curvature less than 8 cm is significant signal degradation observed.”

“While our textiles are robust to a wide range of deformations, excessive degrees of bending and twisting can degrade networking performance. Further work may address this challenge by strategically placing more rigid textiles to limit deformation in the most sensitive regions of the metamaterial while preserving flexibility in regions essential for body motion.”

4. Although soft devices based on textiles are very attractive, they have not been commercialized yet (at least in a mass scale). There can be many issues, and reliable mass production and performance/quality guarantee for long-term customer use would be challenging issues. More discussions in terms of technology transfer to the industry can be added for readers.

Reply: We appreciate the reviewer's insightful comment and suggestions regarding the commercialization potential of our technology. As requested, we have included a discussion regarding the technology transfer aspects in the Discussion section of the revised manuscript. The main points are summarized below:

- As noted by the reviewer, the cost, reliability, manufacturing, and user benefit are important factors that have impeded the mass adoption of electronic textiles. However, we believe that our textiles possess two unique characteristics that may reduce the obstacles to technology transfer. First, they are designed to work with wirelessly-enabled medical implants and enhance their diagnostic or therapeutic capabilities, without directly providing any such function themselves. This design minimizes regulatory barriers for adoption. Second, our textiles are fully passive, which significantly reduces challenges related to manufacturing and reliability compared to fully active electronic textiles.

- We acknowledge that there are still significant gaps that need to be addressed, particularly in the areas of material integration and robustness. Due to the passive nature of the metamaterials, we are optimistic about leveraging industry advances in conductive textiles to address remaining challenges in durability, quality control, and manufacturing.
- We recognize that user adoption is a significant challenge for any new technology, and our metamaterial textiles are no exception. We believe that implant networking presents a particularly promising opportunity for textile-based devices because the potential diagnostic/therapeutic benefits can justify the additional effort required to use them. For instance, these textiles could be used temporarily during a critical period, such as post-surgery recovery, and need not be worn permanently.

We have added a new paragraph in Discussion section.

“The clinical translation of our textile-based networking approach presents challenges in cost, manufacturing, reliability, and user adoption. However, our approach has several advantages that may facilitate adoption by users, including the fully passive nature of our textiles that simplifies manufacturing and enhances reliability. Additionally, our textiles are designed to work with existing wirelessly-enabled implants, which reduces regulatory hurdles. Furthermore, the rapid advances in conductive textiles in the industry can be harnessed to address challenges in cost, durability, and quality control. Ultimately, the success of our networking approach will depend on its ability to provide sufficient diagnostic or therapeutic benefits to justify the effort required to use it. Initial efforts may target applications in which the textiles are worn temporarily during critical periods, such as monitoring during post-surgery recovery. As the technology matures, the technology may be used to target applications that require long-term use and/or have vital health functions, such as closed-loop drug delivery.”

Reviewer #3

“In this work, the authors reported a direct implant-to-implant wireless networking at the scale of the human body using metamaterial textiles. The textiles facilitate non-radiative propagation of radio-frequency signals along the surface of the body. The authors’ group has reported the wireless networking at the scale of the human body using metamaterial textiles (Nature Communications, 2020, 11, 444; Nature Communications, 2022, 13, 2190; Nature Electronics, 2021, 4, 382–39) And the implantable sensor network was also reported in Nature Electronics, 2019, 2, 335–342). All these previous publications largely undermined the novelty of current manuscript. I do not see enough advancement in current work to justify a publication in Nature Communications.”

We appreciate the reviewer's feedback on our work, but respectfully disagree with the reviewer's assessment. While it is true that our group has published articles on metamaterial textiles in the past that can be used with wearable devices, this study represents the first instance of implantable devices networked on a human scale. This is a substantially distinct and noteworthy achievement that required fundamental advances in metamaterial design for manipulating both deep and surface waves, as well as technically challenging large animal experiments. Our proposal to create an implant network using textiles and its successful demonstration *in vivo* should represent a significant enough advance to warrant publication in *Nature Communications*.

Our previous work related to wireless sensor interrogation using circuits operated at exceptional points (*Nature Electronics*, 2019, 2, 335–342) does not bear any technical similarity with this work beyond its thematic link. In particular, our previous work focuses on the readout of microsensors using an external device and does not address the challenge of implant-to-implant networking.

Again, we thank all the reviewers for their constructive feedback. We believe that the manuscript is much improved as a result.

REVIEWERS' COMMENTS

Reviewer #1 (Remarks to the Author):

Authors answered clearly to the reviewer concerns.
New interesting applications and future works can be envisioned.

Reviewer #2 (Remarks to the Author):

All comments from the reviewer were addressed well in the revised manuscript which is now ready for publication.

June 6, 2023

Re: Nature Communications manuscript NCOMMS-23-05814A

To the reviewers,

We thank all the reviewers for accepting our responses to their comments and their contribution on reviewing the paper.

Reviewer #1 (Remarks to the Author):

Authors answered clearly to the reviewer concerns.

New interesting applications and future works can be envisioned.

Reviewer #2 (Remarks to the Author):

All comments from the reviewer were addressed well in the revised manuscript which is now ready for publication.